# Ultra-durable superhydrophobic cellular coatings

Wancheng Gu[1,9], Wanbo Li [2,3,9] ✉, Yu Zhang[1], Yage Xia[1], Qiaoling Wang[1], Wei Wang[1], Ping Liu[1], Xinquan Yu[1], Hui He[4], Caihua Liang[4], Youxue Ban[5], Changwen Mi [5], Sha Yang[6], Wei Liu [6], Miaomiao Cui[2], Xu Deng [7] ✉, Zuankai Wang [2,8] ✉ & Youfa Zhang[1] ✉

Developing versatile, scalable, and durable coatings that resist the accretion of matters (liquid, vapor, and solid phases) in various operating environments is important to industrial applications, yet has proven challenging. Here, we report a cellular coating that imparts liquid-repellence, vapor-imperviousness, and solid-shedding capabilities without the need for complicated structures and fabrication processes. The key lies in designing basic cells consisting of rigid microshells and releasable nanoseeds, which together serve as a rigid shield and a bridge that chemically bonds with matrix and substrate. The durability and strong resistance to accretion of different matters of our cellular coating are evidenced by strong anti-abrasion, enhanced anti-corrosion against saltwater over 1000 h, and maintaining dry in complicated phase change conditions. The cells can be impregnated into diverse matrixes for facile mass production through scalable spraying. Our strategy provides a generic design blueprint for engineering ultra-durable coatings for a wide range of applications.

Engineering ultra-durable coatings that are capable of resisting accretion of matters in diverse phases (liquid, vapor, or solid) and meanwhile endowing multifunctions is essential to numerous practical applications such as aero/marine engineering[1–4], petrochemical engineering[5,6], biology[7], architecture[8], and heat transfer[9–11]. However, it appears mutually exclusive to design one coating that displays all these preferred features through structural design[12]. First, introducing rough structures on coatings is preferred for liquid repellence[13–16], but, which also results in strong local vapor permeability[17], large adhesion with solid particles[18,19], and reduced mechanical strength[12,20]. Second, lowering the surface energy of coatings can decrease the affinity for

accretion of liquid and solid[21–23], such as in the case of superhydrophobic surfaces, which in turn gives rise to limited chemical bonds within the matrix and weak adhesion with the underlying substrate.

Over the past decade, extensive attempts have been made to mitigate these challenges (Supplementary Table 1). Superhydrophobic and durable coatings have been engineered by the delicate choice of the bulk matrix[24–26] with high elastic modulus[27,28], high elasticity[29–31], self-healing capability[32–35], or self-similar structures[16,30]. Despite this, the superior durability of the matrix could not be translated into the entire coating. Alternatively, constructing rigid armors with refined

[1]Jiangsu Key Laboratory of Advanced Metallic Materials, School of Materials Science and Engineering, Southeast University, Nanjing 211189, P. R. China. [2]Department of Mechanical Engineering, City University of Hong Kong, Hong Kong SAR 999077, P. R. China. [3]Interdisciplinary Research Center, School of Mechanical Engineering, Shanghai Jiao Tong University, Shanghai 200240, P. R. China. [4]School of Energy and Environment, Southeast University, Nanjing 210096, P. R. China. [5]School of Civil Engineering, Southeast University, Nanjing 211189, P. R. China. [6]Nano and Heterogeneous Materials Center, School of Materials Science and Engineering, Nanjing University of Science and Technology, Nanjing 210094, P. R. China. [7]Shenzhen Institute for Advanced Study, University of Electronic Science and Technology of China, Shenzhen 518110, P. R. China. [8]Department of Mechanical Engineering, The Hong Kong Polytechnic University, Hong Kong SAR 999077, P. R. China. [9]These authors contributed equally: Wancheng Gu, Wanbo Li. ✉e-mail: wanboli@sjtu.edu.cn; dengxu@uestc.edu.cn; zk.wang@polyu.edu.hk; yfzhang@seu.edu.cn

microstructures on surfaces, including interconnected frames[12], cavities[36–38], and pillars[39–42], improves the overall mechanical durability. However, such a method calls for sophisticated manufacturing[43] and also becomes ineffective in harsh environments involving sharp and localized abrasion and impact. To date, designing and scalable fabrication of a universal, ultra-durable coating that repels multiphase matters remain challenging.

Here we propose a cellular design approach that integrates both structural and functional robustness in one coating. As shown in Fig. 1a, the key lies in designing a cellular unit, or cell, that consists of a rigid microshell and releasable nanoseeds. The cells are mechanochemically controlled to impart the coating's ultra-durability. Mechanically, cells act as a strong shield to protect the surface structures when the applied load is smaller than their critical fracture point. Whereas, at larger loads, the top cells can be broken and nanoseeds are instantaneously released by the shear force, featuring a shear-adaptive release, thus maintaining the water repellence (Supplementary Fig. 1a and Discussion 1). Chemically, we leveraged the heterogeneous chemistry of the cells by fully salinizing the nanoseeds and partially salinizing the shells, which enables the cells to have a strong bonding strength with the matrix, meanwhile keeping a global superhydrophobicity, as probed by Density Functional Theory (DFT) simulation (Supplementary Fig. 1b–d, and Discussion 2). These cells are also well dispersed in various matrixes to engineer cellular coatings with strong mechanical durability and multifunctionality simultaneously.

## Results

### Design and fabrication of cellular coatings

In our design, we chose porous diatomite, silica nanosphere, and epoxy resin as the shell, seed, and matrix, respectively. The cellular coating was prepared in three steps: silanizing shells and seeds, forming cells, and suspending the cells into the matrix. In the first step, the additional amount of siloxane for silanizing the shells and seeds

was respectively controlled to render the cells chemically heterogeneous. Then the cells are formed by impregnating the shells with the seeds using a vigorous stirring in butyl acetate. The as-prepared cells are finally dispersed stably in epoxy, a representative multipurpose matrix thus forming the coating suspension (see Supplementary Methods and Supplementary Fig. 2a for details). The coating suspension can be sprayed to form a covalently bonded coating (Supplementary Figs. 3 and 4) on various substrates (e.g., glass, metal, ceramics, polymer composite materials, paper, sponge, etc.) at 80 °C for 1 h.

Figure 1b shows the scanning electron microscopy (SEM) image of the as-fabricated cell, which possesses a pancake shape with a diameter of 20 μm and a thickness of 3 μm. The cell is porous with the pore size ranging from 50 to 600 nm. Zoom-in SEM inspection reveals that the nanopores are impregnated with a large number of nanoseeds (Supplementary Fig. 5). The maximum load of nanoseeds within the cell is 30 wt.%, as determined by SEM and Brunauer–Emmett–Teller analyses (Supplementary Fig. 6). Under this critical value, the cells can be closely packaged in the epoxy matrix, forming a compact and continuous bulk phase meanwhile keeping nanoscale roughness (Fig. 1c and Supplementary Fig. 7).

As a first glance of the feasibility of our design, we compared the tensile fracture strength $\sigma_c$ and wettability (i.e., water contact angle $\theta^*$ and roll-off angle $\theta_{roll-off}$) of the cellular coatings with those of conventional coatings, all of which were prepared in a similar process as described in Supplementary Methods and Supplementary Fig. 2b–d. As shown in Fig. 1d, for coatings with a water contact angle above ~160°, the fracture strength of the cellular coating is ~10 times of the control samples of shell-alone or seed-alone coating, suggesting the simultaneous preserving of both mechanical strength and water repellency. The tensile fracture strength and wetting property can be tailored and optimized by controlling cell contents $\alpha$ (i.e., the weight ratio of cells in the matrix) and chemical bond density $\beta$, as evidenced

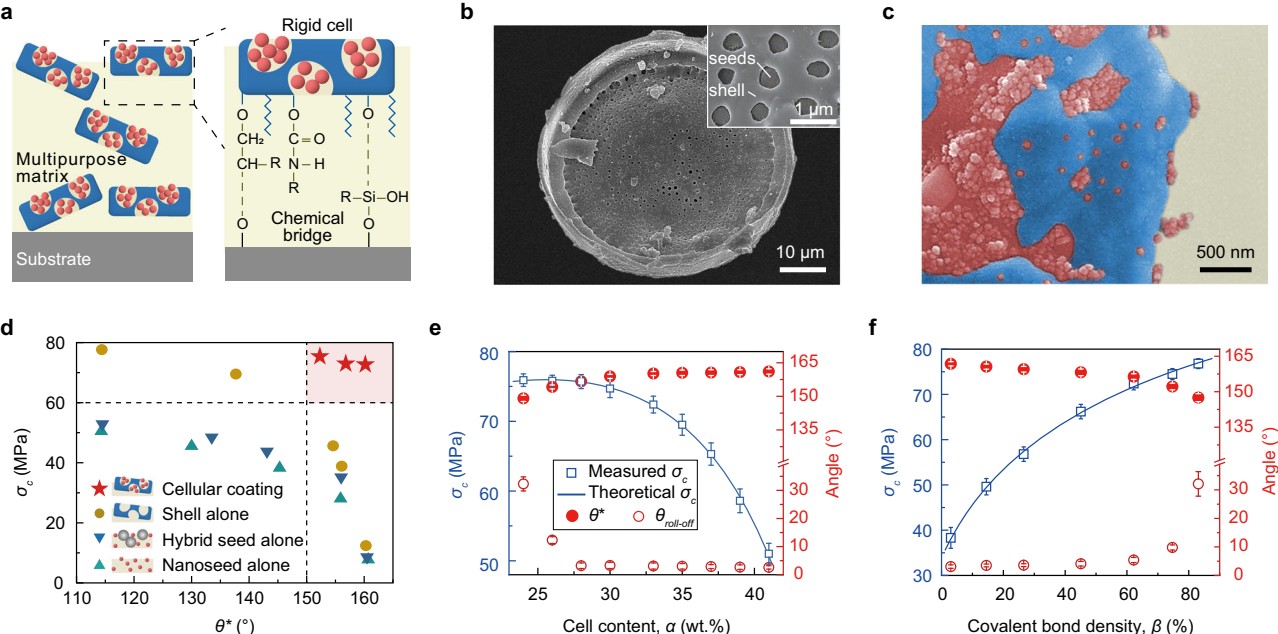

**Fig. 1 | Design and characterization of cellular coatings. a** Schematic illustration of cellular design. The rigid cells serve as a shield to protect the nanoseeds from mechanical abrasion and a chemical bridge to bond with the matrix and substrate. **b** SEM image of the cell. **c** A zoom-in SEM image of the coating surface. The diatomite shell, epoxy matrix, and silica nanoseeds are rendered blue, yellow, and red, respectively. **d** Simultaneous exhibition of high fracture strength $\sigma_c$ and water contact angle $\theta^*$ by cellular coatings, which is in contrast to the tradeoff facing the controls (i.e., coatings with seed alone, shell alone). The horizontal and vertical dotted lines refer to the matrix strength and superhydrophobic boundary, respectively. **e, f** Change of the coating fracture strength $\sigma_c$ and water repellence ($\theta^*$ and $\theta_{roll-off}$) of cellular coatings as a function of the cell content $\alpha$ (**e**) and chemical bond density $\beta$ (**f**). The errors represent the standard deviations from at least three independent experiments.

in Fig. 1e, f. The reinforcement in both the tensile fracture strength and wettability rendered by the cellular design is also applicable to other kinds of the matrix such as acrylic, polyurethane (PU), and ceramic (Supplementary Fig. 8).

Notably, the reinforcement effect of basic cells and chemical bond density can be predicted by theoretical models based on the Griffith−Irwin−Orowan theory[44] and DFT simulation. Briefly, the fracture strength $\sigma_c$ is expressed as:

$$\sigma_C^2 \propto -\alpha^3 + A\alpha^2 + B\alpha \qquad (1)$$

where $\alpha$ is cell content, A and B are the coefficients related to the crack length and elastic modulus of cell and matrix (see Supplementary Discussion 3 and Supplementary Fig. 9a for details), and here we assume the initial cracks start from the interface between a silanized cell and the matrix. Similarly, the effect of covalent bond density $\beta$ on the fracture strength $\sigma_c$ follows:

$$\sigma_C^2 \propto \frac{(\beta + C)^2}{\beta} \qquad (2)$$

where C is a correction parameter that reflects the heterogeneity of covalent bonds and cell distribution (see Supplementary Discussion 3 and Supplementary Fig. 9b for details). Besides, we further analyzed the influence of the cell size on coating strength (see Supplementary Discussion 3 for details). The theoretical values predicted by our models fit our experimental results as shown in Fig. 1e, f, and Supplementary Fig. 10.

## Mechanical properties

We then performed the microscopic test to demonstrate the reinforcement effect of the basic cells. Figure 2a plots the load-displacement curve for the individual cell in nanoindentation testing. A breakpoint occurs at 3.2 mN, corresponding to the onset of cell fracture. Once infused into coatings, the basic cells serve as a mechanical shield that withstands the major stress and protects the nanoseeds against mechanical scratch under loads below 3.2 mN. In sharp contrast, without using the basic cells, as manifested by the nanoseed-alone coatings, the coatings were easily pierced through and even scratched off from the substrate (Fig. 2a, b). Compared to those control samples (i.e., shell-alone, hybrid-seed-alone, and nanoseed-alone coatings), the cellular coating exhibits a remarkable enhancement in both the hardness and elastic modulus (Fig. 2c). When further increasing the load beyond a critical force, we observed the fracture of the cell, which led to the instantaneous release of the stored nanoseeds over the damaged region and thereby maintaining the surface roughness (Fig. 2d−f and Supplementary Figs. 11 and 12).

## Ultra-durability

We then investigated the mechanical durability of the cellular coatings against Taber abrasion and jet impalement, respectively. We found the cellular coating could tolerate 1000 abrasion cycles under a 1-kg load (Fig. 3a, b, and Supplementary Fig. 13 and Movie 1) regardless of the matrixes used, suggesting that the reinforcement effect mainly originates from the cells rather than the matrix. In contrast, all the control samples including the nanoseed-alone, shell-alone, hybrid seed-alone coating, matrix-alone coating, and a commercial superhydrophobic coating were worn away and lost water repellence after tens of abrasion cycles, as shown by the insets in Fig. 3b and Supplementary Fig. 14. The advantages of the cellular coatings over the control samples and the state-of-art coatings are shown in Fig. 3c and Supplementary Fig. 15, in which we plot the critical failure load and wearing coefficient (defined as the maximum abrasion cycles that the coating can tolerate per thickness unit without the loss of water repellence,

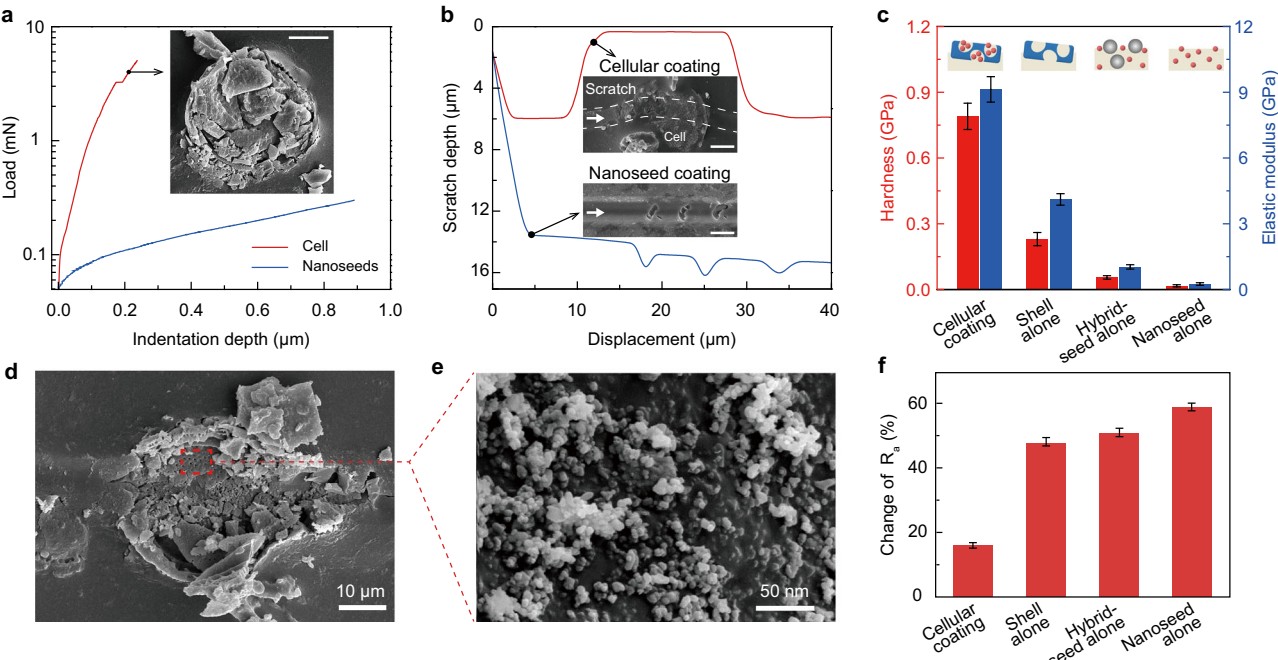

**Fig. 2 | Mechanical properties of the cell. a** Mechanical characterization of the cell (in red) and nanoseeds (in blue) by nanoindentation. The apparent breakpoint shows the fracture strength of the cell. The inset shows the SEM image of the fractured cell under 4-mN load. Scale bar: 10 μm. **b** Variation of the scratch depth as a function of the indenter displacement under 0.4-mN load, in which the red line refers to the cellular coating and the blue line refers to the nanoseed-alone coating. As shown in the inset images, the cellular coating was kept almost intact whereas a deep groove was created on the nanoseed-alone coating by the indenter. Scale bar: 10 μm. **c** Hardness and elastic modulus of different coatings. The errors represent the standard deviations from at least three independent experiments. **d, e** SEM images of the cellular coating (**d**) and zoom-in view of the released nanoseeds (**e**) after micro-scratching under a load of 4 mN. **f,** Comparison of roughness change of different coatings after abrasion. The errors represent the standard deviations from at least three independent experiments.

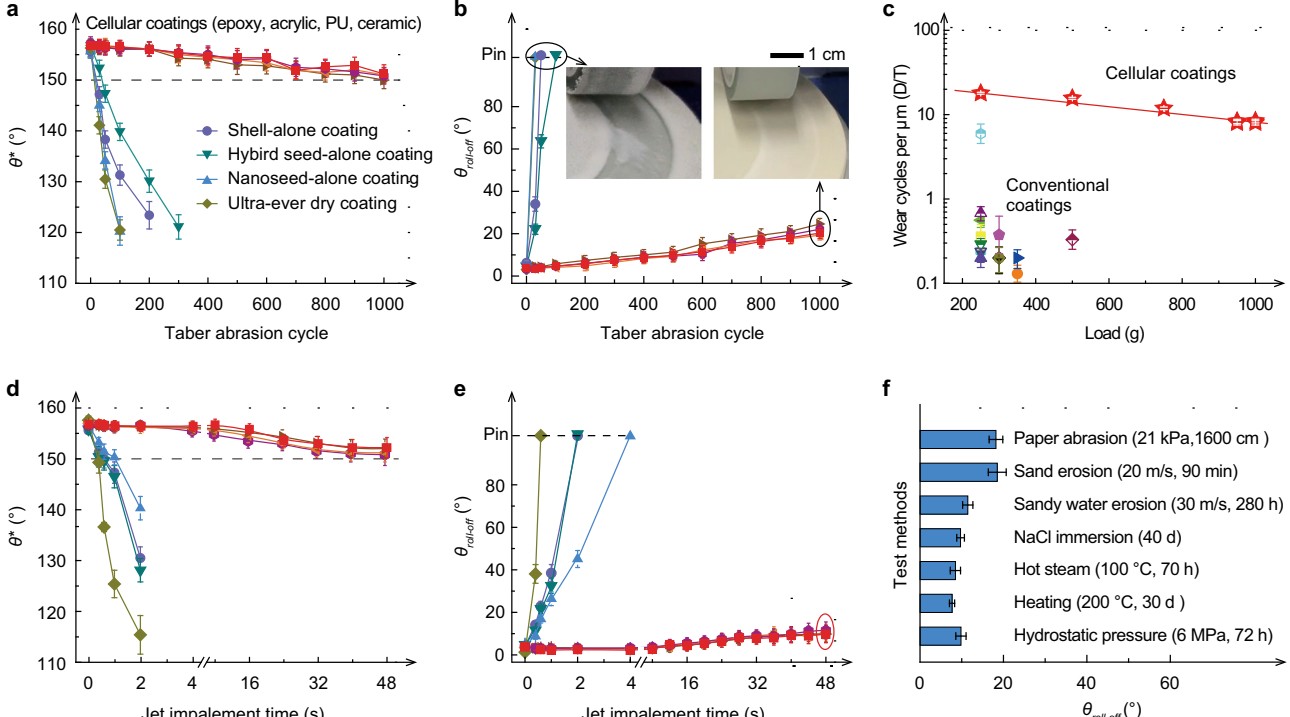

**Fig. 3 | Mechanical robustness of cellular coatings (coating thickness ~80 μm).**
**a, b** Evolution of water contact angles (**a**) and roll-off angles (**b**) of different coatings during Taber abrasion under 1-kg load. For comparative study, cellular coatings in different matrixes (i.e., epoxy, acrylic, polyurethane, and ceramic), shell-alone coating, hybrid seeds-alone coating, nanoseed-alone coating, and Ultra-ever Dry coating were also tested. The inset image in (**b**) highlights the distinct worn scars on the cellular coating and the other coatings after abrasion. **c** Comparison of the wear resistance of the cellular coating with that of the existing reports. See Supplementary Fig. 15 for details. **d, e** Evolution of water contact angles (**d**) and roll-off angles (**e**) of different coatings along with the water jet impact time (jet velocity ~40 m s⁻¹, Weber number ~44,444). **f** Summary of the roll-off angles on the cellular coatings after diverse standard durability tests. The errors represent the standard deviations from at least three independent experiments.

see Supplementary Methods for details). The wear resistance of the cellular coating is 30–100 times higher than that of its conventional counterparts. Figure 3d, e show the variation of $\theta^*$ and $\theta_{\text{roll-off}}$ of different coatings during water jet impalement at *We* of ~44,444, where $We = \rho v^2 l/\gamma$ is the Weber number, with $\rho$, $v$, $l$, and $\gamma$ being the density, impact velocity, characteristic length, and surface tension of the jet fluid, respectively. Notably, the cellular coating can withstand impalement for 48 s, whereas all the control coatings lost superhydrophobicity in only 2 s (Supplementary Fig. 16). The cellular coating can also timely regain its superhydrophobicity by a gentle abrasion that allows the release of the stored nanoseeds (Supplementary Fig. 17 and Movie 2). Even after diverse abrasion and crush (Supplementary Figs. 18 and 19), repeated human stepping (Supplementary Movie 3), tape-peeling, and high-pressure hydrostatic immersion (Supplementary Fig. 20), chemical, thermal, and aging treatments (Supplementary Fig. 21), and substrate adhesion tests (Supplementary Fig. 22), the cellular coatings show a negligible drop in the water repellence, suggesting the superior stability of our cellular coating (Fig. 3f and Supplementary Table 2).

**Multiphase repellence**
More importantly, we show that the coating reinforced by the cellular design exhibits strong repellence to vapor, liquid, and solid matters and even during multi-phase changes, which is crucial for marine, building, and energy applications. Before the tests, all the coatings are treated with severe mechanical damage (e.g., 100-cycle sandpaper abrasion) as described in the method. First, the cellular coating possesses a strong barrier against corrosive vapor and condensates due to the vapor impermeability (Supplementary Fig. 23) and strong liquid repellency (Supplementary Fig. 24). Moreover, even after 50-day vapor

treatment, no pitting corrosion was observed on the steel plate covered by the cellular coating, whereas the steels covered by the nanoseed-alone coating and the matrix-alone hydrophobic coating were corroded within several days (Fig. 4a). Even after immersion in seawater for 150 days, the impedance of the cellular coating is maintained at ~10⁹–10¹⁰ Ω cm², which is four orders of magnitude larger than those of conventional coatings as plotted in Fig. 4b, suggesting its high energy barrier against liquid corrosive (Supplementary Fig. 25). Second, the treated cellular coating demonstrates a strong repellence to solid phases, including mortar and ice. The mortar with a size ranging from 1 μm to 10 cm in both precursor and solidified states can slide away from the tilted coating owing to its ultra-low adhesion – less than 0.1 kPa (Fig. 4c, and Supplementary Fig. 26a–c and Movie 4). The cellular coatings also show an ultralow ice adhesion of ~20 kPa, which is lower than the critical adhesion for passive ice removal by wind or vibration (Supplementary Fig. 26d). Third, the cellular coating on the heat exchanger can remarkably improve heat-transfer efficiency in diverse phase-change processes. We show that the coating suppresses frosting (Supplementary Fig. 27) and facilitates frost self-peeling off (Fig. 4d and Supplementary Movie 5) at low temperatures, saving more than ~65% energy compared to a commercial hydrophilic coating (Fig. 4e). In other complex phase-change processes involving condensation, dirt passive removal, frosting, and defrosting (Fig. 4f and Supplementary Fig. 28), the cellular coating also demonstrates generic energy-saving performance.

In summary, the cellular design we proposed resolves the contradictory requirements on structure, chemistry, and surface/bulk phase properties and enables an ultra-durability of repellent coatings. All the materials used are commercially available and environmentally friendly, and the fabrication is also scalable (Supplementary Fig. 29 and

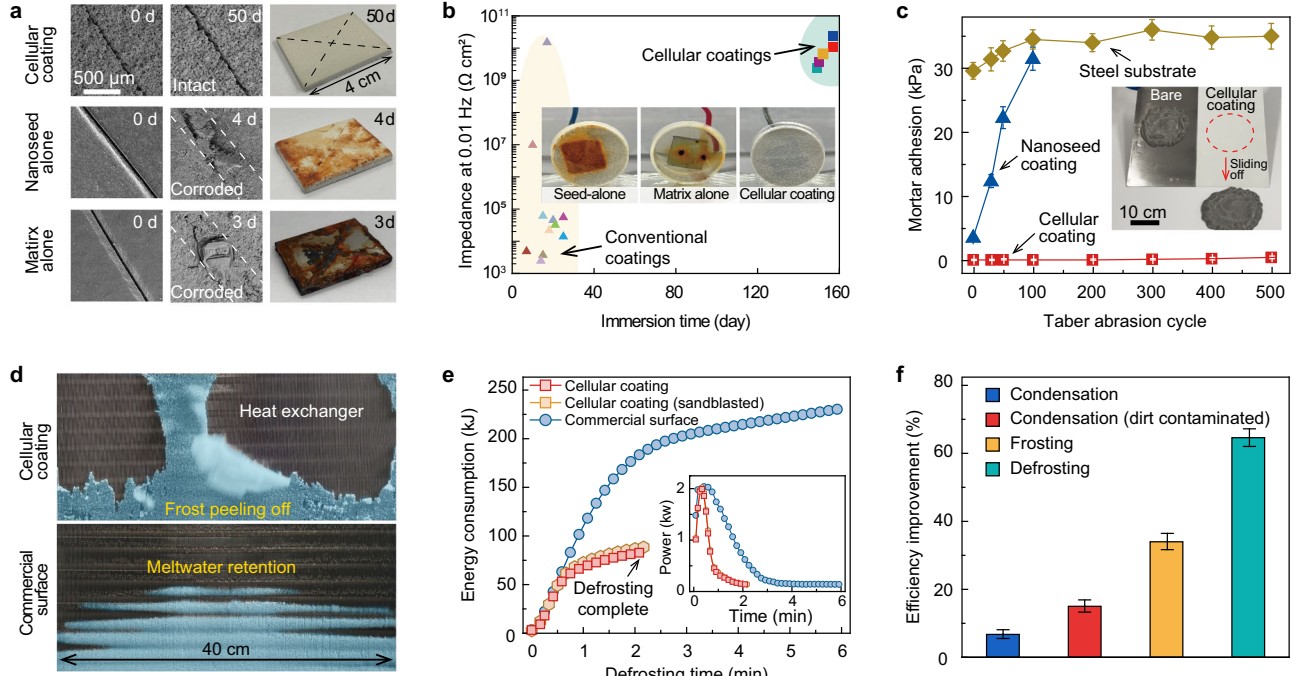

**Fig. 4 | Multiphase repellence of cellular coatings for sustainable applications (coating thickness ~80 μm). a, b** Anti-corrosion. **a** SEM and optical images of the Q235 steel plates with the cellular coatings and control coating during salt spraying corrosion. Note the cross notch on the coating surfaces. **b** Comparison of the corrosion resistance in 3.5 wt.% NaCl solution of the cellular coatings with the conventional coatings. The insets are the optical images of the coatings after 60-day immersion. See Supplementary Fig. 25 for details. **c** Anti mortar adhesion. Evolution of mortar adhesion with different coatings during Taber abrasion under 1-kg load. The inset image shows the mortar self-removal from the cellular coating at a tilt angle of ~30°, which highlights the solid phase resistance of the cellular

coating. The errors represent the standard deviations from at least five independent experiments. **d–f** Thermal management through anti-frosting. For comparison, the cellular coating after 10-min sandblasting and a commercial hydrophilic coating were tested. **d** Defrosting behaviors on heat-exchangers with the cellular coating and commercial hydrophilic coating. The frost sheet is rendered blue. **e** Energy consumption of the heat exchangers with different coatings during defrosting. Inset shows the relationship between the defrosting power and time. **f** Heat-transfer improvement by cellular coatings in comparison with the commercial hydrophilic coating. The errors represent the standard deviations from at least three independent experiments.

Supplementary Table 3). Nonetheless, we further demonstrated the ultra-durability of the cellular coating made of fluorinated substances (Supplementary Fig. 30). We envision that the cellular coating is also promising for many other real-world applications such as moisture-adsorption proof, drag reduction, anti-fouling, radiative cooling, and energy harvesting[45–47].

## Methods

### Materials
Absolute ethanol and butyl acetate were purchased from Sigma Aldrich, USA. Silica nanospheres (average size: 13 nm) were purchased from Nissanchem, Japan. Ammonia hydroxide (28%) and silica microspheres (average size: 10 μm) were purchased from Aladdin, China. Diatomite (~5–50 μm), octyltriethoxysilane (OTS), bisphenol A epoxy resin, polyacrylic acid resin, polyurethane resin (PU), and polyphenylsiloxane matrix (ceramic) were all purchased from Wanqing Corporation, China.

### Coating preparation
The coating suspension was prepared in three steps: silanizing the diatomite and nanosilica, preparing cells by impregnating diatomites with nanosilica, and suspending the cells into the matrix. For silanization, 36 g of silica nanospheres were first dispersed in a solution (containing 800 mL of ethanol, 80 mL of deionized water, and 40 mL of ammonia hydroxide) and mechanically stirred for 5 min, and then 4.8 mL of OTS was added to the dispersion and stirred at 50 °C for 24 h to silanize the silica surface. Finally, the silica nanospheres were freeze-dried in a vacuum environment (~0.04 MPa) at 30 °C for 24 h. The diatomite shells were also

silanized using this method, except in which the volume of OTS was adjusted to 0.6 mL, 1.2 mL, 1.6 mL, 3.6 mL, and 4.8 mL for tailoring the surface chemistry. For cell preparation, the resultant silica nanospheres (0.6 g) were loaded within the diatomite shell (2 g) by stirring their mixture in butyl acetate (15 g) for 10 min. Finally, the cells were suspended into the epoxy matrix (6.8 g) to form a homogeneous dispersion which kept stable for long-term storage (at least for 6 months) under room conditions. The coating suspension could be applied to various substrates (e.g., glass, metal, ceramics, polymer composite materials, paper, sponge, etc.) through spraying, brushing, and dipping, and then drying at 80 °C for 1 h. The control samples, including coatings containing either shell alone, hybrid seed alone, or nanoseed alone, were also prepared in similar procedures with respective modifications illustrated in Supplementary Fig. 2.

### Surface morphologies
The morphologies of the coatings were characterized by scanning electron microscopy (SEM, Nova Nano SEM450, USA) at an accelerating voltage of 15 kV and transmission electron microscopy (TEM, Talos F200X, Thermo Fisher Scientific, USA) at a beam acceleration of 200 kV.

### Characterization of surface chemical groups
The surface groups and the coating composition were analyzed by Fourier transform infrared spectroscopy (FTIR, Nicolet iS10, Thermo Fisher Scientific, USA). The sample was first dried at 100 °C for 24 h, and its FTIR spectrum was recorded using an ATR model.

## Bulk mechanical strength measurements

The mechanical strength of the coating bulk was evaluated by a material tensile test (CMT5105, MTS System Corporation, China) according to the ASTM D638 standard. In brief, the sample was cast in a mold with a standard dumbbell shape. The narrow region of the dumbbell is 5-mm thick, 13-mm wide, and 57-mm long. Then, the samples were fixed on the MTS instrument and stretched at a rate of 5 mm min$^{-1}$. The stress at which the fracture occurred was defined as the fracture strength at the break.

## Water repellence measurements

The water repellence of the coating was assessed by measuring the static contact angle ($\theta^*$) and roll-off angle ($\theta_{roll-off}$) of water droplets using a contact angle meter (OCA 15Pro, Dataphysics, Germany). For static contact angle measurement, a 5-μL water drop was placed on the coating surface and reached static. The contact line of the rounded water drop with the coating surface was captured and analyzed by software (SCA20-Software for OCA and PCA). For roll-off angle measurement, a 10-μL water drop was placed on the horizontal coating surface and reached static. Then, the coating surface was tilted at a rate of 0.5° per second to allow the droplet to roll off. The tilt angle at which the droplet started rolling was denoted as the roll-off angle.

## Microscratching

For the microscratching test, a single cell was immobilized on a silicon surface by the epoxy matrix. Then the cell was loaded on an instrument (Nano Test Vantage, Micro Materials Corporation, UK) with a diamond indenter (tip size: 5 μm), and the indenter scratched over the samples at a velocity of 5 μm s$^{-1}$ under different normal loads (0.4 and 4 mN). The scratch morphology and depth profile were characterized for evaluating the cell's mechanical strength.

## Nanoindentation

The cell was loaded on Nano Test Vantage (Micro Materials Corporation, UK) and an indentation was created on the sample surface by using a diamond indenter (tip size: 50 nm) with a constantly increased load at a rate of 0.025 mN s$^{-1}$ to 0.5 mN. The load-depth curve, elasticity modulus, and hardness (Oliver-Pharr method) were recorded to evaluate the mechanical strength of the cell. The elasticity modulus of the matrix was also tested using the same method.

## Taber abrasion

The Taber abrasion test (ASTM D4060 standard) was performed to evaluate the wear resistance of the coatings. All the coatings (thickness ~80 μm) were spray-coated on circular glass plates with a diameter of ~100 mm, and were worn by two abrasive wheels (Calibrase® CS–10, TABER INDUSTRIES, USA) under different loads, 250, 500, 750, and 1000 g. Each rotation of the sample plate was counted as one abrasion cycle. During the test, $\theta^*$ and $\theta_{roll-off}$ were measured to assess the water repellence of the coating after each 25 abrasion cycles. For quantifying the wear resistance of the superhydrophobic coating, a wear index $W$ is defined as follows,

$$W = D/T \tag{3}$$

where, $D$ is the maximum abrasion cycles that the coating can tolerate without the loss of water repellence, and $T$ is the coating thickness (μm).

## High-speed jet impact

The resistance to high-speed jet damage was evaluated by the water-jet impact test. The samples were all prepared on rec rectangular glass slide (75 mm × 25 mm) with a coating thickness of ~80 μm. Note that, the samples for all the following durability tests possessed the same size, otherwise specially noted. Briefly, for the water-jet impact test,

water was loaded in a syringe connected to a compressive nitrogen cylinder, and was pneumatically ejected out at a pre-defined pressure value. The speed of the water was determined by the jet motion video captured by a high-speed camera. For each test cycle, 500-mL water was ejected out within 4 s at a speed of ~40 m s$^{-1}$ (Weber number, $W$ ~44,444), and both $\theta^*$ and $\theta_{roll-off}$ were measured for assessing the water repellence.

## RCA abrasion

The coating wear resistance was also assessed according to ASTM F2357-04 standard by using RCA-7-IBB abrader (Biuged, China), as shown in Supplementary Fig. 18a. The coating surface was continuously abraded by the abrasive paper (width ~0.6875 inches) under a load of 12 kPa. Both $\theta^*$ and $\theta_{roll-off}$ were collected after each 160-cm abrasion distance for the water repellence assessment.

## Sandpaper abrasion

The sandpaper abrasion test (Grit No. 240, Electro coated aluminum oxide waterproof abrasive paper, Diamond Brand, China) was performed according to the procedure reported by Yao Lu et al.[1] Both $\theta^*$ and $\theta_{roll-off}$ were measured after every 40 cycles of sandpaper abrasion for the evaluation (Supplementary Fig. 18d).

## Sandblasting

The sandblasting test was conducted according to the GJB 150.12 A standard, as shown in Supplementary Fig. 19a. The setup consisted of three modules, a high-speed airflow generator, sand loader, and sample holder in a closed-loop chamber. After the sample was placed on the holder tilted at an angle of 10°, the sands continuously impacted the sample surface at a relative density of 2.2 g m$^{-3}$ and a flow rate of 20 m s$^{-1}$ driven by the high-speed wind. During the test, both $\theta^*$ and $\theta_{roll-off}$ were measured after each 10-min duration for assessing the water repellence.

## Falling sand impact

The resistance to mechanical impact was evaluated by the sand impact test. During the test, the silica sand (size ~100–250 μm) fell at a rate of 40 g min$^{-1}$ from a height of 30 cm, and impacted the coating surface at a tilt angle of 45 degrees (Supplementary Fig. 19c). Both $\theta^*$ and $\theta_{roll-off}$ were measured after every 5 min of sand impact for the evaluation.

## Sandy water erosion

Sandy water erosion was performed for mimicking ship movement in the water with sand particles. For this test, the sample was fixed in a baker with the coating surface immersed in sandy water (SiO$_2$ sand: 100–250 μm, sand content: 4 g L$^{-1}$). The sandy water was stirred by a mechanical agitator at a speed of 5000 r min$^{-1}$. The linear shear speed of flow was ~ 30 m s$^{-1}$, which is approximately two times higher than the fastest speed of an Aircraft Carrier (15.3 m s$^{-1}$). After each 20 h test, $\theta^*$ and $\theta_{roll-off}$ were measured to evaluate superhydrophobicity.

## Tape-peeling

Tape-peeling tests (ASTM D3359-17 standard) were performed to assess the adhesion strength of the coating. For the test, the coating surface was first mounted with a 3M™ VHB tape (width ~2 cm, adhesion value ~ 3000 N m$^{-1}$), and then was pressed by rolling over a copper rod (weight ~4 kg). After the tape was peeled off, both $\theta^*$ and $\theta_{roll-off}$ were measured for the water repellence assessment (Supplementary Fig. 20b).

## High hydrostatic pressure treatment

Coatings operating underwater face high hydrostatic pressure which might drive the failure of superhydrophobicity. To demonstrate such feasibility, we tested the coating in a homemade high-pressure vessel which consists of a pressure pump, vessel cavity, and pressure control

valve, as shown in Supplementary Fig. 20c. The coating was first immersed in the 3.5 wt.% salt water and then was transferred into the vessel cavity with a pressure of 6 MPa. This pressure is corresponding to the deepwater pressure under a depth of ~612 m. During the test, both $\theta^*$ and $\theta_{roll-off}$ were recorded after each 12 h treatment.

## Chemical stability tests
The chemical stability was tested by directly immersing the coatings into chemical solutions, i.e., 3.5 wt.% NaCl solution, HCl solution (pH-5), and NaOH solution (pH-9). After each 5-day immersion, both $\theta^*$ and $\theta_{roll-off}$ of the coatings were recorded.

## Thermal stability tests
The thermal stability was tested by placing the coating in liquid nitrogen (−196 °C) and oven (350 °C), respectively. The thermal treatment duration at each temperature was 2 h. After that, both $\theta^*$ and $\theta_{roll-off}$ were measured to assess the superhydrophobicity. For long-term thermal stability, we kept the sample in the oven at 200 °C for 40 days, during which, the water repellence was evaluated after every 5 days when the sample was cooled down to room temperature.

## Anti-corrosion test
The anti-corrosion performances of the coatings on Q235 steel substrates were evaluated through a neutral salt spray test and salt solution immersion. The long-term stability against neutral salt spray was tested according to the ASTM B117 standard. The samples were prepared according to the following procedure. First, the Q235 steel plates (size, 40 mm × 30 mm × 3 mm) were ultrasonicated in ethanol absolute to clean the oil contamination and polished with sandpaper to remove the oxide layer. After that, a thin layer of coating (thickness ~ 80 μm) was prepared on the steel surface via the spray method. Finally, the cured coating on the Q235 steel substrate was cross scratched with a blade to expose the underlying Q235 steel and the sample was ready for salt spray test. For the test, the sample was fixed at a tilt angle of 70° in a salt spray chamber, in which the environmental temperature was 35 °C and the concentration of salt solution for the spray was 5 wt.%. During the salt spraying, the surface morphology and the coating superhydrophobicity were recorded after each 12 h to evaluate the anti-corrosion capability.

The anti-corrosion capability was further quantified by the electrochemical impedance spectroscopy when the sample was immersed in the salt solution. For the sample preparation, the wired Q235 steel substrate (10 × 10 × 10 mm) was first sealed with epoxy resin and then was sequentially polished by sandpapers with different roughness ranging from 150, 600, 800, 1200, to 2000 grit, until the steel substrate was exposed and smoothed. Finally, a thin layer of coating (thickness ~80 μm) was sprayed onto the Q235 steel substrate. To obtain the impedance and open circuit potential, the as-prepared samples were immersed in the 3.5 wt.% salt solution and were characterized by an electrochemical workstation (CHI660E, CH Instruments, Inc., China) using a three-electrode system, where the Q235 steel with coating, a platinum sheet, and a Hg/HgCl$_2$ electrode saturated with KCl (+0.2415 V vs. SHE) were used as the working electrode, counter electrode, and reference electrode, respectively. For the electrochemical impedance measurement, a sinusoidal signal was applied with an amplitude of 50 mV at a frequency ranging from $10^{-2}$ to $10^5$ Hz.

## Mortar adhesion
The adhesion of mortar at solid and viscous states was tested. The solid mortar adhesion was tested on a tinplate plate (30 cm × 30 cm) with half of the surface covered by cellular coating and the other half kept bare for comparative study. The mortar slurry (weight ~150 g, and cement/sand/water mass ratio 2:6:1) was poured onto the coating and bare surface of the substrate. After solidifying in room conditions

(25 °C) for 24 h, the tinplate plate was tilted at ~30° to enable the solid mortar to self-remove. The mortar adhesion strength was measured according to the scheme shown in Supplementary Fig. 26a. In brief, the mortar confined in stainless steel cubes was applied to the substrates. After solidifying for 24 h at room temperature, the dry mortar was detached using a biaxial motion stage at a velocity of 80 μm s$^{-1}$ with a gauge monitoring the force. The peak force was recorded as the mortar adhesion.

To demonstrate the anti-adhesion of viscous mortar, A mortar slurry drop (100 μL) on the coating surface was compressed and then released to allow the mortar drop to bead up. The compression and decompression procedure was captured by a stereomicroscope. The roll-off behavior of the mortar drop was determined by slowly tilting the coating surface, the angle at which the drop started rolling off was denoted as its roll-off angle. To demonstrate the anti-adhesion of mortar slurry, the coating surface was mechanically damaged sequentially by steel-wool abrasion (average force: 7.6 N), sandpaper abrasion (Grit No. 240, average force: 6.6 N), and screwdriver scratch (average force: 4.8 N), then the mortar slurry was poured on and kept under 4-kPa pressure for 20 min. Finally, the sample was tilted to allow the mortar slurry to roll off.

## Ice adhesion
The ice adhesion was tested using a similar method to the mortar adhesion test. First, pure water in stainless-steel cubes (10 mm × 10 mm × 30 mm) was applied in contact with the coating at −20 °C. After freezing for 1 h, the ice was detached by a biaxial motion stage at a velocity of 80 μm s$^{-1}$, with a gauge monitoring the force. The maximum force for completely removing was recorded as the ice adhesion strength of the sample. For a comparative study, the ice adhesion of control coatings and cellular coatings after Taber abrasion under a 1-kg load were also measured.

## Frosting and defrosting
The ultra-low-temperature frosting test was performed in a cooling chamber with the coating temperature (−20 °C), air temperature, and air relative humidity being precisely controlled. Before the test, the coatings were mechanically damaged by Taber abrasion for 200 cycles under a 1-kg load. For defrosting, the coating temperature was tuned to 25 °C in 30 s, and the defrosting behavior was captured by a camera and stereomicroscope. Besides, the duration of frosting and defrosting, as well as the energy consumption of defrosting were analyzed for evaluation.

## Hot vapor condensation
To evaluate the stability of the coating in a highly humid and hot environment, vapor condensation was conducted. The vapor flow from boiling water was directly pumped onto a vertically placed coating. The behavior of nucleation, growth, and spontaneous removal of condensate water droplets was captured by the camera for evaluation of the superhydrophobicity. The coatings after Taber abrasion (200 cycles under a 1-kg load) were also used for comparative analysis.

## Low-temperature condensation
For the low-temperature condensation test, the coating surface was tilted at an angle of ~ 45°, the temperatures of the coating surface and air were −2 °C and −25 °C, respectively, and the relative humidity of the air was ~70%. The condensation behaviors before and after Taber abrasion (200 cycles under a 1-kg load) were recorded using a high-speed camera (FASRCAM Mini UX700, Photron, Japan) at a rate of 6000 frames per second and a stereomicroscope (Zoom 6000, Navitar, USA). Besides, the surface coverage of the condensates was analyzed for evaluation (Supplementary Fig. 24d).

## Moisture adsorption

The moisture adsorption test was performed in a sealed chamber according to the ASTM B117 standard. The samples were fixed at a tilt angle of 70° in the chamber with an ambient temperature of 35 °C and relative humidity of ~90%. The moisture absorption rate, defined as the ratio between the weight increment and the initial weight of the coating, was measured after each 12 h to evaluate the moisture adsorption.

## Heat transfer test

The heat transfer performance of heat exchangers with cellular coating was evaluated according to the GB 50736 standard in an enthalpy difference laboratory which can simulate real-world conditions, for condensation, frosting, and defrosting. For the test, the heat exchanger was placed in an air duct, in which, the airflow speed, humidity, and air temperature can be precisely controlled and monitored. The temperature of the exchanger was also controlled by the introduction of a liquid cold source or heat source (i.e., glycol/water mixture or pure water with the respective specific heat capacity of 3.38 and 4.2 J g$^{-1}$ °C$^{-1}$) with constant temperatures at a fixed flow speed of 25 g min$^{-1}$. The temperature of the liquid source in the inlet and outlet of the heat exchanger were both measured by using thermocouples for calculating the heat transfer quantity.

For condensation, the temperatures for the cold source (i.e., glycol/water mixture) in inlet and air flows were 5 and 35 °C, respectively. The relative air humidity was 70%. For frosting, the temperatures for the cold source (i.e., glycol/water mixture) in the inlet and airflow were −15 and −2 °C, respectively, and the air relative humidity was 80%. After frost formation, the cold source was changed to the heat source (i.e., water) at 30 °C for the defrosting test.

With the above results, the energy conversion $Q_{ex}$ of the heat exchanger is calculated as follows:

$$Q_{ex} = c_{so}m_{so}(t_{so.in} - t_{so.out}) \tag{4}$$

where $c_{so}$ is the specific heat capacity of the cold fluid, $m_{so}$ is the flow rate of the cold fluid, and $t_{so.in}$ and $t_{so.out}$ are the inlet temperature and outlet temperature of the cold fluid.

Thus, the total energy conversion during the process of frosting/condensation can be expressed as:

$$Q_{total} = \int_0^t Q_{ex}dt \tag{5}$$

## Data availability

The data supporting the conclusions of this study are included in the article and the supplementary information files.

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

## Acknowledgements

We acknowledge financial support from the National Natural Science Foundation of China (grant nos. 52071076 by Youfa.Z. and 22072014 by X.D.), the Sichuan Outstanding Young Scholars Foundation (21JCQN0235 by X.D.), the Innovation and Technology Fund (No. GHP/021/19SZ by Z.W.), and the Shanghai Pujiang Program 22PJ1406100 by W.L.

## Author contributions

Youfa.Z. and Z.W. conceived the research. Yooufa.Z., Z.W., X.D., and W.L. supervised the research. W.G., W.L., and X.Y. designed the experiments. Yu.Z., W.W., and P.L. participated in the discussion of the experimental design. W.G., W.L., and Q.W. prepared and characterized the coatings. W.G., W.L., Q.W., Yu.Z., Y.X., M.C. assembled, characterized the samples and carried out the experiments. H.H. and C.L. conducted the heat transfer test. Y.B. and C.M. built theoretical models with input from Yo.Z. S.Y. and W.L. performed the Finite Element and Density Functional Theory simulations with input from Yo.Z. All authors analysed the data. Youfa. Z., Z.W., W.L., X.D., and W.G. wrote the manuscript with input from the other authors.

## Competing interests

Youfa.Z., W.G., Yu.Z., and X. Y. are co-authors on a US patent application (no. 17834820), which was filed on Jun. 07, 2022 and described the methods used herein. The remaining authors declare no competing interests.
