## [Peer Review File · Nature Communications]

Ultra-durable superhydrophobic cellular coatingsEditorial Note: This manuscript has been previously reviewed at another journal that is not operating a transparent peer review scheme. This document only contains reviewer comments and rebuttal letters for versions considered at Nature Communications.

REVIEWER COMMENTS

Reviewer #1 (Remarks to the Author):

In this manuscript, the authors reported a cellular coating that imparts liquid-repellence, vapor-imperviousness, and solid-shedding capabilities without the need for complicated structures and fabrication processes. The durability and strong resistance to accretion of different matters of our cellular coating are evidenced by strong anti-abrasion, enhanced anti-corrosion against saltwater over 1000 h, and maintaining dry in complicated phase change conditions. However, the manuscript lacked remarkable innovation because some similar component systems have been studied extensively (Advanced Functional Materials 2022, 32, 2113297.; Nature communications 2021, 12, 982;). So, I suggest the authors further emphasize the highlights and distinguishing features of the manuscript. I believe this manuscript could be considered after solving below additional concerns to meet the high impact requirements of Nature communications.

1. It will be better if the authors could provide more detail preparation process of the superhydrophobic cell coating.
2. Why did the authors use bisphenol A epoxy resin in the manuscript. In addition, as we all known, the organosilanes have so much variety. So why did the authors use octyltriethoxysilane rather than other silanes? Please provide more clearly reasons.
3. In Equation 6, E_1 , and E_2 are the elasticity modulus of cells, and matrix. How obtain the E_1 and E_2 values? Or how to test the two values?
4. The authors provided theoretical models for optimizing the coating strength in supporting information, so could you educe the relationship between the coating microstructure (e.g., particle and cell diameters, particle and cell distributions, and quantities) and the coating strength? I think the theoretical models have a deeper impact on the superhydrophobic coating.

Reviewer #2 (Remarks to the Author):

This work propose a cellular coating design approach. The coating is mainly composed of micro-sized porous diatomite shell, releasable silica nanosphere seed and multipurpose matrix. Meanwhile, finite element (FE) modeling of physical shield, simulation of chemical bridge and theoretical models for optimizing the coating strength are also provided. The experimental test results show that the cellular coatings have strong mechanical durability, good superhydrophobicity, substrate adhesion and strong repellence to vapor, liquid, and solid matters. The approach of the coating design is new and more

creative. This work provides a new feasible way for engineering ultra-durable coatings. However, some issues should be addressed after the review.

1. Although the authors demonstrated the excellent performance of the coating through experimental tests, the necessary theoretical analysis and discussion were missing for each performance test result. Such as these results of the tensile fracture strength, the mechanical durability, strong repellence to vapor, liquid, and solid matters, ice adhesion, heat-transfer efficiency, etc. Similarly, there are few analysis and discussion on the calculated results of these theoretical models. For example, it is pointed out that the covalent bonds are formed between coatings and various substrates. How the covalent bonds are formed and the relationship between the covalent bonds and the performance of the coating have not been properly discussed and analyzed. These theoretical analysis and discussion are very important for understanding this kind of coating.

2. In terms of the comparison of test results, especially the mechanical friction test, mechanical brushing test and substrate adhesion test, the author only conducted the comparison of the test results under different conditions for their own experiments, and did not compare these results reported by others. This will lead to a lack of rationality in comparison results. These comparison are important for the illustration of the coating performance.

Based on the above problems, I don't think this work can be published in Nature Communications.

Reviewer #3 (Remarks to the Author):

The paper presents an interesting experimental study on the fabrication of durable superhydrophobic surfaces, with an extended characterization of durability in a variety of conditions.

I think the study is interesting and worth publishing in this journal. However, I have some comments that the authors should address to improve the paper clarity.

1. I am not sure why the authors call the diatomite “shell” and silica “nanoseeds”. Looking at the SEM in Fig. 1c, it seems that the silica nanoparticles are decorating the diatomite external surface. I thus do not really agree with the authors’ claim: “Zoom-in SEM inspection reveals that the nanopores are impregnated with a large number of nanoseeds” (page 4). It looks to me that diatomite and silica are creating particles with dual scale roughness, which is known to be helpful with to enhance hydrophobicity. Can the authors provide better images to support their statement?

2. What are the wetting properties of the epoxy matrix? I was not able to find the information in the paper or in the SI. In case the epoxy is hydrophilic, was it hydrophobized to ensure good adhesion between the matrix and the so-called cell? If not, how is good adhesion possible?

3. Calling the roll of angle θ_r (Figure S12 to S19) is a bit misleading, as this is normally the symbol used to indicate the receding contact angle. I suggest changing the symbol.

Reviewer #1:

In this manuscript, the authors reported a cellular coating that imparts liquid-repellence, vapor-imperviousness, and solid-shedding capabilities without the need for complicated structures and fabrication processes. The durability and strong resistance to accretion of different matters of our cellular coating are evidenced by strong anti-abrasion, enhanced anti-corrosion against saltwater over 1000 h, and maintaining dry in complicated phase change conditions. However, the manuscript lacked remarkable innovation because some similar component systems have been studied extensively (*Advanced Functional Materials* 2022, 32, 2113297.; *Nature communications* 2021, 12, 982;). So, I suggest the authors further emphasize the highlights and distinguishing features of the manuscript. I believe this manuscript could be considered after solving below additional concerns to meet the high impact requirements of *Nature communications*.

Response: We thank the reviewer for reviewing and commenting on our manuscript, and allowing us to emphasize the novelty. The key innovation of our work lies in designing cells that consist of rigid microshells and releasable nanoseeds. The cells were mechanochemically controlled to impart the coating ultra-durability. Mechanically, cells act as a strong shield to protect the surface structures when the applied load is smaller than their critical fracture point. Whereas, at larger loads, the top cells can be broken and nanoseeds are instantaneously released by the shear force, featuring a shear-adaptive release, thus maintaining the water repellence. Chemically, we leveraged on the heterogeneous chemistry of the cells by fully salinizing the nanoseeds and partially salinizing the shells, which enables the cells to have a strong bonding strength with the matrix, meanwhile keeping a global superhydrophobicity. We have added these statements in the revised manuscript. Please kindly see Lines 40-50 in the manuscript.

The cellular design is distinct from those referred to by the reviewer. The work in *Adv. Funct. Mater.*, 2022, reported a strategy of introducing high-strength materials and adhesives to improve the mechanical durability of coatings. The dual-scale particles were used to create a structural hierarchy to enlarge the water repellency. Zhang's work (*Nat. Commun.* 2021, 12, 982) used a self-similar structure design to improve durability, that is, the water repellency was maintained by exposing underlying similar structures after abrasion. Although previous studies also used chemical components (e.g., silica) similar to our work, the mechanical shield or heterogeneous chemistry was not involved,

a core innovation of our method. Moreover, compared with these strategies, our cellular design achieved 2-3 orders enhancement in the abrasion resistance, as characterized by the remarkable improvement of the wear coefficient, as we have presented in Fig. 3c in the manuscript. We have reviewed the mechanism of these reports in the revised manuscript and added them to the reference list. Please see Lines 30-33 and References 16 and 26.

Comment 1. It will be better if the authors could provide more detail preparation process of the superhydrophobic cell coating.

Response: Thanks for the valuable suggestion. We have added the detailed description of the preparation process in the revised manuscript. Please kindly see Lines 53-63.

Comment 2. Why did the authors use bisphenol A epoxy resin in the manuscript. In addition, as we all known, the organosilanes have so much variety. So why did the authors use octyltriethoxysilane rather than other silanes? Please provide more clearly reasons.

Response: Thank you for the comments. First, the criterion for choosing the matrix is the capability of adhering to a wide range of substrates. According to this, our cellular coatings could be prepared by various matrixes, including epoxy resin, polyurethane resin, polyacrylic acid resin, and ceramic matrix, which all obtained similar ultra-durability and multiphase repellence (Please see Lines 81-83 and Fig. 3 in the manuscript, and Figs. S8, S25 and S30 in the Supplementary information). In the current manuscript, epoxy resin was used to represent common matrix systems, which facilitates expanding the application of cellular coatings in different fields, i.e., maintaining the original coating systems in these fields.

Second, the criterion for choosing silanes is their reaction activity with -OH. A moderate activity allows us to control the density of the modified silane group on cells without forming sol-gels. We used octyltriethoxysilane as a representative because its terminal ethoxy groups are much more stable compared with some other groups, for example, the chlorine group of n-octadecyltrichlorosilane. The ethoxy groups need a certain amount of water to facilitate the hydrolysis reaction and form the hydroxyl groups at a relatively low speed. Then, the octyltriethoxysilane could graft onto the silica particles by dehydration condensation reaction to provide low surface energy.

Comment 3. In Equation 6, E_1 , and E_2 are the elasticity modulus of cells, and matrix. How obtain the E_1 and E_2 values? Or how to test the two values?

Response: We thank the reviewer for the thoughtful comments. The elasticity modulus of cells and matrix was directly obtained by the nanoindentation tests. We have added the methods in the revised version. Please kindly see Section 4.2 in the revised Supplementary information.

Comment 4. The authors provided theoretical models for optimizing the coating strength in supporting information, so could you educe the relationship between the coating microstructure (e.g., particle and cell diameters, particle and cell distributions, and quantities) and the coating strength? I think the theoretical models have a deeper impact on the superhydrophobic coating.

Response: Thank you for the helpful suggestions. In the previous modeling, we have studied the influence of cell content on the coating strength. The cell distribution is one of the parameters that determines the heterogeneity of covalent bonds in the coating. However, since the cell distribution and heterogeneity of covalent bonds are both hard to characterize using the existing methods, we used a parameter C for correction. Please see Equation 2 in the manuscript (Lines 93-94).

Here, we further analyzed the influence of the cell size on coating strength as follows. The mechanical strength of the cellular coating with different cell diameters d (here, d is the average diameter characterized by laser particle size analyzer, Mastersizer 3000, UK) can also be predicted based on the Griffith-Irwin-Orowan theory, which can be expressed as the following equation:

$$\sigma_c = \left(\frac{2E\gamma_p}{\pi a}\right)^{1/2} \quad (1)$$

where E is the elasticity modulus of the coating, γ_p is the total plastic work before the coating breaking, and a is the crack length. The elasticity modulus of composite coatings can be represented by a generalized rule of the form as below:

$$E = E_1V_1 + E_2V_2 \quad (2)$$

where E , E_1 , and E_2 are the elasticity modulus of coating, cells, and matrix, respectively, and V , V_1 and V_2 are the volume fraction of coating, cells, and matrix, respectively. Then, V_1 and V_2 can be respectively expressed as follow:

$$V_1 = \frac{\pi(d_1^2h_1+d_2^2h_2+d_3^2h_3+\dots+d_N^2h_N)}{4V} = \frac{\pi(i_1d_1^3+i_2d_2^3+i_3d_3^3+\dots+i_Nd_N^3)}{4V} \quad (3)$$

$$V_2 = 1 - \frac{\pi(i_1 d_1^3 + i_2 d_2^3 + i_3 d_3^3 + \dots + i_N d_N^3)}{4V} \quad (4)$$

where $d_1, d_2, d_3 \dots d_N$ are the diameter of each cell, $h_1, h_2, h_3 \dots h_N$ are the height of each cell, and $i_1, i_2, i_3 \dots i_N$ are the correlation coefficient between diameter and height, respectively. Therefore, Eq. (2) can be deduced as:

$$\begin{aligned} E &= E_1 \frac{\pi(i_1 d_1^3 + i_2 d_2^3 + i_3 d_3^3 + \dots + i_N d_N^3)}{4V} + E_2 \left(1 - \frac{\pi(i_1 d_1^3 + i_2 d_2^3 + i_3 d_3^3 + \dots + i_N d_N^3)}{4V}\right) \\ &= E_2 + \frac{\pi}{4V} (E_1 - E_2) (i_1 d_1^3 + i_2 d_2^3 + i_3 d_3^3 + \dots + i_N d_N^3) \\ &= E_2 + \frac{\pi}{4V} (E_1 - E_2) (i_1 j_1^3 + i_2 j_2^3 + i_3 j_3^3 + \dots + i_N j_N^3) d^3 \\ &= D + M d^3 \end{aligned} \quad (5)$$

where $j_1, j_2, j_3 \dots j_N$ are the correlation coefficient between the diameter of each cell and the average diameter. As the initial crack is the interface between the cell and matrix, the crack length a is in direct proportion to the cell diameter d :

$$a = k_3 d \quad (6)$$

By substituting Eq. (5) and (6) into Eq. (1), the relationship between the coating fracture strength and cell diameter d can be expressed as:

$$\sigma_c^2 = \frac{2\gamma_p}{\pi k_3 d} (D + M d^3) \quad (7)$$

that is:

$$\sigma_c^2 \propto P d^2 + \frac{Z}{d} \quad (8)$$

where parameters P and Z are the contribution factors illustrating the reinforcement effect of the cell and the deterioration effect induced by the initial crack in the interface of the cell and matrix, respectively. In other words, when the cell diameter is relatively small, the reinforcement effect of the cell plays a dominant role. On the contrary, when the cell diameter is relatively large, the deterioration effect of the crack determines the coating strength.

The fit of Eq. (8) with the experimental data points in Fig. R1, demonstrated a high consistency between the theoretical calculation and experimental results, suggesting the feasibility of the Griffith-Irwin-Orowan theory for predicting the mechanical strength of superhydrophobic coatings.

We have added the corresponding discussion and Fig. R1 to the revised version. Please see Lines 94-97 in the revised manuscript, and Fig. S10 and Supplementary Text 3 (Pages 14-15) in the revised

Supplementary information.

Fig. R1. Change of the coating fracture strength σ_c as a function of the cell diameter d .

Reviewer #2:

This work propose a cellular coating design approach. The coating is mainly composed of micro-sized porous diatomite shell, releasable silica nanosphere seed and multipurpose matrix. Meanwhile, finite element (FE) modeling of physical shield, simulation of chemical bridge and theoretical models for optimizing the coating strength are also provided. The experimental test results show that the cellular coatings have strong mechanical durability, good superhydrophobicity, substrate adhesion and strong repellence to vapor, liquid, and solid matters. The approach of the coating design is new and more creative. This work provides a new feasible way for engineering ultra-durable coatings. However, some issues should be addressed after the review.

Response: We thank the reviewer for reading and commenting on our work and also appreciating the novelty and impact of our work. The comments are addressed point-by-point below.

Comment 1. Although the authors demonstrated the excellent performance of the coating through experimental tests, the necessary theoretical analysis and discussion were missing for each performance test result. Such as these results of the tensile fracture strength, the mechanical durability, strong repellence to vapor, liquid, and solid matters, ice adhesion, heat-transfer efficiency, etc. Similarly, there are few analysis and discussion on the calculated results of these theoretical models. For example, it is pointed out that the covalent bonds are formed between coatings and various substrates. How the covalent bonds are formed and the relationship between the covalent bonds and the performance of the coating have not been properly discussed and analyzed. These theoretical analysis and discussion are very important for understanding this kind of coating.

Response: Thank you for the critical suggestions. First, in our previous manuscript, we have built models to analyze the relationship between coating parameters (e.g., cell content and covalent bond density) and the coating strength. Further, as suggested by Reviewer 1, we have analyzed the influence of cell size on the coating strength, please see the response to Comment 4 in Pages 3-5 of this response letter. All the theoretical modeling has been validated by the experiments. Please kindly see Fig. 1 in the manuscript and Fig. R1 in the response letter. The cellular design and theoretical modeling allow us to develop superhydrophobic coatings with strong mechanical strength and a compact and continuous bulk phase, which are the basis for achieving ultra-durability and functional performance.

However, to our best knowledge, these performances can only be tested according to the industrial standards or common test procedures in the literature.

Second, the covalent bonds formed through the condensation reactions between the -OH group on the cell surface and the reactive groups of the matrix (e.g., epoxy groups and isocyanate groups), as expressed in Fig. R2. The covalent bond density is one main factor influencing the coating strength, which we have theoretically and experimentally analyzed in the manuscript. Please see Lines 78-80 and 91-97, and Fig. 1f.

We have added Fig. R2 to the revised Supplementary information as Fig. S3. The corresponding discussion is provided in the revised manuscript (Lines 60-63).

Fig. R2. Chemical reaction for forming chemical bonds between matrix and cells.

Comment 2. In terms of the comparison of test results, especially the mechanical friction test, mechanical brushing test and substrate adhesion test, the author only conducted the comparison of the test results under different conditions for their own experiments, and did not compare these results reported by others. This will lead to a lack of rationality in comparison results. These comparison are important for the illustration of the coating performance.

Based on the above problems, I don't think this work can be published in Nature Communications.

Response: We thank the reviewer for the constructive suggestions. In the previous manuscript, we have evaluated the abrasion resistance (characterized by the wearing coefficient) of the cellular coating and existing state-of-art works. Please see Fig. 3c in the manuscript. Some other comparisons of the durability were presented in Table S1 in the Supplementary information (please see Pages 46-47 in SI). Here, for quick catching of the comparison, we summarize some key results (sandpaper abrasion, jet impalement, falling sand impact, and substrate adhesion) in the Table below.

Table R1. Comparison of durability with the reported work

Durability test	Reported work	This work	Improvement (fold)
Sandpaper abrasion	Bearing 40 cycles (Science 2015, 347, 1132) Bearing 100 cycles (J. Mater. Chem. A 2017, 5, 19297) Bearing 10 cycles (Mater. Horiz. 2019, 6, 1057) Bearing 30 cycles (J. Mater. Chem. A 2017, 5, 14542) Bearing 50 cycles (Chem. Eng. J. 2020, 392, 124834) Bearing 1000 cm (Adv. Funct. Mater. 2022, 32, 2113297)	Bearing 400 cycles, i.e., 8000 cm	4-40
Jet impalement	Bearing 2-s impact (Nat. Mater. 2018, 17, 355) Bearing 400-mL impact (Nature 2020, 582, 55)	Bearing 48-s impact, i.e., 6000-mL impact	15-24
Falling sand impact	Bearing 20-g impact (Science 2012, 335, 67) Bearing 30-g impact (Chem. Eng. J. 2019, 373, 298) Bearing 15-min impact (Adv. Funct. Mater. 2017, 27, 1604261)	Bearing 60-min impact, i.e., 2400-g impact	4-120
Substrate adhesion test	Bearing 52-time tape-peeling (ACS Nano 2017, 11, 1113) Bearing 10-time tape-peeling (J. Mater. Chem. A 2016, 4, 4107) Bearing 80-time tape-peeling (Chem. Eng. J. 2017, 322, 10) Bearing 30-time tape-peeling (Nat. Mater. 2018, 17, 355) Bearing 100-time tape-peeling (J. Mater. Chem. A 2020, 8, 3509)	Bearing 200-time tape-peeling	2-20

We are delighted by the reviewer's words, "the approach of the coating design is new and more creative" and "this work provides a new feasible way for engineering ultra-durable coatings" in the overall comment. In the current manuscript, we have further emphasized the novelty of our work and improved the theoretical modeling and comparison with the state-of-the-art reports. We hope the reviewer finds our work meets the standard of *Nature Communications*.

Reviewer #3:

The paper presents an interesting experimental study on the fabrication of durable superhydrophobic surfaces, with an extended characterization of durability in a variety of conditions. I think the study is interesting and worth publishing in this journal. However, I have some comments that the authors should address to improve the paper clarity.

Response: We thank the reviewer for reading and commenting on our work. We are delighted that the reviewer finds our work interesting and recommends publication. The comments are addressed point-by-point below.

Comment 1. I am not sure why the authors call the diatomite "shell" and silica "nanoseeds". Looking at the SEM in Fig. 1c, it seems that the silica nanoparticles are decorating the diatomite external surface. I thus do not really agree with the authors' claim: "Zoom-in SEM inspection reveals that the nanopores are impregnated with a large number of nanoseeds" (page 4). It looks to me that diatomite and silica are creating particles with dual scale roughness, which is known to be helpful with to enhance hydrophobicity. Can the authors provide better images to support their statement?

Response: We thank the reviewer for the helpful suggestion. We provide a new figure to show the cell better. As shown in the figure below (Fig. R3), the seeds were all impregnated in the shell. We have added the figure as the inset of Fig. 1b in the revised manuscript and Fig. S5 in the revised Supplementary information. The corresponding discussion is provided in the revised manuscript (Lines 66-67).

Fig. R3. Morphologies of the nanoseeds in the shell.

Comment 2. What are the wetting properties of the epoxy matrix? I was not able to find the information in the paper or in the SI. In case the epoxy is hydrophilic, was it hydrophobized to ensure

good adhesion between the matrix and the so-called cell? If not, how is good adhesion possible?

Response: Thank you for the thoughtful comments. We used hydrophilic epoxy as one representative matrix. Please see the detail in the response to Comment 2 by Reviewer 1. The strong adhesion was achieved by the covalent bonding reaction between the -OH group on the cell surface and the reactive group (e.g., epoxy groups and isocyanate groups) of the epoxy matrix (Fig. R4). The bond formation after each reaction process was demonstrated by the FTIR spectra in Fig. R5.

Fig. R4. Chemical reaction for forming chemical bonds between matrix and cells.

Fig. R5. Chemical bond characterization. FTIR spectra of the silanized shells, impregnated cells after silanization, and cured cellular coating. The partially silanized cell surface presented absorption peaks for -OH and -CH_x groups, which kept stable after the cells were dispersed in the epoxy matrix. After curing, the remaining -OH groups on the cell thoroughly reacted with active groups in the matrix, forming strong covalent bonds.

Comment 3. Calling the roll of angle theta_r (Figure S12 to S19) is a bit misleading, as this is normally the symbol used to indicate the receding contact angle. I suggest changing the symbol.

Response: Thanks for the suggestion. We have replaced all the symbols (θ_r) of the roll-off angle with the new symbol ($\theta_{roll-off}$). Please kindly see Figs. 1 and 3 in the revised manuscript, and Figs. S14, S17-21, S23, S24, and S30 in the revised Supplementary information.

REVIEWERS' COMMENTS

Reviewer #1 (Remarks to the Author):

In the revised manuscript, the authors further emphasized and summarized the highlights and distinguishing features. Moreover, the influence of the cell size on coating strength has been added. In terms of the novelty and importance, the revised manuscript could be accepted in Nature communications.

Reviewer #2 (Remarks to the Author):

The author has answered the questions raised, and the problems existing in the paper have been revised. Therefore, the decision to accept is given.

Reviewer #3 (Remarks to the Author):

The authors have addressed the issues I had raised. As such, my suggestion is now to publish the paper.

Reviewer #1:

In the revised manuscript, the authors further emphasized and summarized the highlights and distinguishing features. Moreover, the influence of the cell size on coating strength has been added. In terms of the novelty and importance, the revised manuscript could be accepted in Nature communications.

Response: We thank the reviewer for reviewing and commenting on our manuscript, and accepting our manuscript in Nature Communications.

Reviewer #2:

The author has answered the questions raised, and the problems existing in the paper have been revised. Therefore, the decision to accept is given.

Response: We thank the reviewer for reading and commenting on our work and also appreciating the decision to accept.

Reviewer #3:

The authors have addressed the issues I had raised. As such, my suggestion is now to publish the paper.

Response: We thank the reviewer for reading and commenting on our work. We are delighted that the reviewer recommends publication.